# Recycled Excavation Soils as Sustainable Supplementary Cementitious Materials: Kaolinite Content and Performance Implications

**DOI:** 10.3390/ma17102289

**Published:** 2024-05-12

**Authors:** Li Ling, Jindong Yang, Wanqiong Yao, Feng Xing, Hongfang Sun, Yali Li

**Affiliations:** 1Key Laboratory of Coastal Urban Resilient Infrastructures (Shenzhen University), Ministry of Education, Guangdong Provincial Key Laboratory of Durability for Marine Civil Engineering, College of Civil and Transportation Engineering, Shenzhen University, Shenzhen 518060, China; popwii90@gmail.com (L.L.); xingf@szu.edu.cn (F.X.); 2Centre for Smart Infrastructure and Digital Construction, School of Engineering, Swinburne University of Technology, Hawthorn, VIC 3122, Australia; 3Research and Development Center of Transport Industry of New Materials, Technologies Application for Highway Construction and Maintenance of Offshore Areas, Fujian Communications Planning & Design Institute Co., Ltd., Fuzhou 350004, China; yangjindong2024@gmail.com; 4State Key Laboratory of Subtropical Building Science, South China University of Technology, Guangzhou 510640, China; 202110181497@mail.scut.edu.cn

**Keywords:** excavation soil, calcined soil, portland cement, kaolinite, metakaolin

## Abstract

In response to the environmental implications of the massive quantities of excavation soil generated by global urbanization and infrastructure development, recent research efforts have explored the repurposing of calcined excavation soils as sustainable supplementary cementitious materials (SCMs). As it is still at an early stage, current research lacks systematic analysis across diverse soil deposits regarding their reactivity and mechanical properties within cementitious binders, despite recognized geographical variability in kaolinite content. Through comprehensive experimentation with soils sourced from four major southern Chinese cities, this study presents a pioneering assessment of the compressive strength, pozzolanic reactivity (X-ray diffraction, Fourier-transform infrared spectroscopy, solid-state nuclear magnetic resonance), and microstructural development (mercury intrusion porosimetry, scanning electron microscopy) of mortars modified by various calcined excavation soils (up to 28 days curing). The experimental data suggest that soils with a kaolinite content above 53.39% produce mortars of equal or superior quality to plain cement mixes, primarily due to their refined pore structures, microstructural densification, and enhanced hydration reactions. The findings highlight kaolinite—specifically, aluminum content—as the principal indicator of excavation soil viability for SCM application, suggesting a promising avenue for sustainable construction practices.

## 1. Introduction

Global urbanization and expansive infrastructure development have led to massive quantities of excavation soil. It is estimated that in urban centers like Shenzhen, China, the annual yield of construction soil waste exceeds 100 million cubic meters [1]. When disposed of improperly, such waste has serious environmental consequences, such as causing landslides, soil erosion, and ecological degradation [2,3].

Soils derived from excavation processes are characterized by the presence of clay minerals, such as kaolinite, montmorillonite, and illite [4,5]. Upon thermal treatment at temperatures ranging from 650–850 °C, kaolinite undergoes a transformation into highly reactive metakaolin (MK) [6,7]. MK can serve as a supplementary cementitious material (SCM), replacing 10–30 wt% of cement in concrete with concurrent enhanced concrete strength and durability, which leads to a reduced carbon footprint of related industries [8,9,10,11,12]. Therefore, excavation soils with high kaolinite content have huge potential for sustainable construction. Additionally, the economic and environmental benefits of using calcined excavation soils as SCMs cannot be ignored; its low-temperature production from construction waste boasts low costs, and its reuse avoids the economic and environmental penalties associated with excavation soil disposal [13].

The potential of kaolinite-rich soils to act as sustainable alternatives to cement has been recognized. R Fernandez et al. [14] highlighted the potential of using calcined soils as reactive mineral admixtures in concrete, emphasizing their ability to lower costs and environmental impacts while enhancing mechanical properties and durability. Samet et al. [15] studied the use of calcined kaolinitic clay sourced from Tabarka (Tunisia) as a pozzolanic material in blended cement (30% replacement), and thermal treatment temperature and fineness were found to affect the compressive strength, with optimal results obtained at 700 °C ground to 7700 cm^2^/g Blaine fineness. Singh et al. [16] examined the production of MK from Indian kaolinitic clays and its effect on Portland cement mortars, finding improved compressive strength, reduced porosity, and identified major hydraulic products like C-S-H and C_4_AH_13_, while durability tests in sulfate solutions showed better strength with 10% MK. Alujas et al. [17] studied the pozzolanic reactivity of Cuban clays calcined at 800 °C, showing that reactivity correlated with structural hydroxyl groups and Al_2_O_3_ content, highlighting the potential of kaolinite-rich red clay soils for obtaining highly reactive pozzolanic materials. Qian et al. [18] explored the utilization of calcined clay in limestone calcined clay cement (LC^3^) production and discovered that the core-shell structure of the calcined clay promoted denser hydration products, leading to significant strength improvements of up to 90% and 137% at 7 and 28 days, respectively. Zhou et al. [19] studied waste London clay as an SCM, finding that calcined at 900 °C, it effectively replaced 30% of CEM I in concrete, achieving similar strengths to traditional additives while reducing CO_2_ emissions by 27%. Despite these promising research findings, the drawback of focusing only on soil from a single source in these studies limited their contribution to the large-scale application of calcined excavation soil as SCM due to the geographical variability in kaolinite content and other mineral compositions in excavated soil [20,21]. Compositions, particularly kaolinite content, in excavation soils demonstrate considerable geographical variation, and their subsequent influence on the reactivity, mechanical characteristics, and microstructure of the resulting cementitious binders remains underexplored [22,23,24,25,26,27]. 

This study investigates the viability of repurposing calcined excavation soils as sustainable SCMs. Excavation soils were sourced from four major cities in southern China and subjected to calcination for use as SCMs. Various characterization techniques were adopted to evaluate the mechanical properties, pozzolanic reactivity, and microstructural evolution of mortars containing the calcined excavation soils. The findings provide critical insights to optimize the recycling of excavation soils with varied mineralogical compositions as high-performance SCMs. Such insights are essential to maximize the reuse potential of excavation soils, reducing environmental impacts and advancing sustainable construction.

## 2. Materials and Methods

### 2.1. Materials

Portland cement (Type P.I 42.5R), produced by the China United Cement Group Co., Ltd. (Beijing, China) was used, containing ≥ 70% tricalcium silicate with a lime–silica ratio ≥ 2.0 (Table 1). Standard sand (Standard of ISO 679:2009 [28], spanning 0.08–2 mm) from Xiamen was procured. Four soils were sourced from different urban infrastructure projects within Shenzhen, Huizhou, Dongguan, and Guangzhou, identified as Soil 1, Soil 2, Soil 3, and Soil 4, respectively (Figure 1). Soil 1 and Soil 3, classified as ultisol, were sourced from locations adjacent to lakes in the Longgang District of Shenzhen and the Huiyang District of Dongguan, respectively. Soil 2, a residual soil, was obtained from Dalang Town in Huizhou, where the soil is notably harder and surrounded by rocky hills. Soil 4, also a residual but softer soil, came from a location near a hillside in the Baiyun District of Guangzhou city. Commercial metakaolin (CMK, 96.65% metakaolin) was sourced from Gongyi, Henan, China.

### 2.2. Cement Mortar and Paste Preparation

Given that kaolinite is predominantly found in fine soil particles, specifically in the size range of 0.3–100 μm, excavation soils were sieved to obtain the portion with an improved concentration of kaolinite [29], leaving the coarser portion (>150 μm), consisting mainly of inert sand and gravel, to be used as aggregates in concrete [30,31,32,33]. The sieved excavation soils were then calcined at 800 °C for two hours to produce metakaolin [34,35,36,37]. As depicted in Figure 2, calcined excavation soil suitable for use as SCMs was obtained through this process.

Mortar and paste samples were prepared by substituting 15% cement with CMK or calcined soil at a 0.5:1 water–binder mass ratio and 3:1 sand–binder mass ratio (only in mortars). To ensure proper mixing, water was first introduced to the mixer. Subsequently, the pre-mixed dry components, OPC, and calcined excavation soil were gradually added while maintaining continuous mixing. Once fully incorporated, sand was introduced. Mixing was conducted at 100 rpm for 30 s, then at 300 rpm for an additional 90 s to ensure uniformity. Then, the mixes were loaded into molds (40 × 40 × 40 mm^3^), cured at a temperature of 20 ± 2 °C and relative humidity (RH) > 95%, and demolded after 24 h. After that, they were cured under the same conditions until testing at 3, 7, and 28 days. 

Samples prepared solely with OPC were designated with the prefixes ‘M0’ and ‘P0’, representing the control mortar and paste samples, respectively. Samples where 15% of the OPC was replaced with CMK carried the prefix “MC” or “PC”. Samples in which 15% of the OPC was substituted with calcined soil used the prefixes “M1” to “M4” or “P1” to “P4”, according to the specific soil types used in the preparation from Soil 1 to Soil 4. 

### 2.3. Methods

The mineralogy of the sieved excavation soils was analyzed using X-ray diffraction (XRD), thermogravimetric analysis (TGA), and X-ray fluorescence (XRF) technologies. The mechanical properties of the mortars were characterized by compressive strength, according to ASTM C109 standards [38]. The microstructure of the mortars was analyzed using D4404 mercury intrusion porosimetry (MIP) and backscattered electron (BSE) technology. The chemical composition and microstructural properties of the resulting hydration products were tested using XRD, Fourier-transform infrared spectroscopy (FTIR), scanning electron microscopy (SEM), and solid-state nuclear magnetic resonance (ssNMR) [39,40,41,42,43,44] with cement pastes.

XRD analyses were performed on the chemical compositions of excavation soil and cement pastes using a Bruker D8 Advance diffractometer (Karlsruhe, Germany) equipped with a Cu Kα radiation source operating at 40 kV and 40 mA. Finely ground powder samples were scanned from 10° to 70° 2θ with a step size of 0.02° and scan speed of 6°/min. Zincite was used as an internal standard at 20% mass ratio [45].

The soil’s phase composition and thermal behavior were examined for TGA and DTG using a NETZSCH STA409PC analyzer (Selb, Germany) with 85 μL Al_2_O_3_ crucibles. Samples weighing approximately 30 mg were heated from 40 °C to 1000 °C at 10 °C/min under nitrogen flow (50 mL/min).

The chemical composition of the soils was quantified by XRF spectroscopy via a SPECTRO iQ II analyzer (Kleve, Germany), featuring a high-resolution silicon drift detector with 145 eV resolution at 10,000 pulses. Samples were prepared as fused beads and analyzed under a vacuum at 25–50 kV and 0.5–1.0 mA for 300 s dwell time.

The compressive strength of the mortar cubes was tested according to ASTM C109 standards using a YZH-300.10 loading frame (Guangzhou, China) with 300 kN capacity at a controlled loading rate of 2.4 kN/s. Three cubes were tested for each mixture at set curing ages.

The pore size distribution of the mortar was measured by employing MIP technology. A Micromeritics AutoPore IV 9500 porosimeter (Norcross, GA, USA), which boasted a maximum pressure capacity of 414 Mpa, was utilized. Prior to filling with mercury, the samples underwent evacuation to reach a vacuum of 50 μm Hg. Subsequently, the samples were subjected to pressurization ranging from 0.10 to 30,000 psi.

The microstructures of the mortar samples were analyzed using BSE technology (Mumbai, India). A Phenom Pro desktop scanning electron microscope (SEM) equipped with a backscattered electron detector from Phenom World (Eindhoven, The Netherlands) was utilized, operating at an acceleration voltage of 10 kV with a standard beam current.

The functional groups of the hydration products in cement paste were examined by FTIR using a PerkinElmer Spectrum One spectrometer (Waltham, MA, USA) equipped with a mid-IR source and DTGS detector. The powder samples were mixed with KBr at a 1:100 ratio, ground manually with an agate mortar and pestle, and pressed into 13 mm diameter pellets under 10 tons of pressure. Transmittance spectra were acquired from 4000 to 400 cm^−1^ at 4 cm^−1^ resolution and 32 scans.

The morphology of hydration products in cementitious materials with calcined soils as additives was examined using a Quanta 250 field-emission SEM produced by FEI (Oregon, USA). Operating in high vacuum mode with an acceleration voltage of 10 kV, the equipment allowed for detailed imaging of the samples. Prior to observation, the bulk samples were carefully fractured to reveal fresh surfaces. Additionally, a gold coating was applied to the samples before they were introduced into the SEM chamber, ensuring the formation of a conductive film for optimal imaging results.

ssNMR spectroscopy was performed on a Bruker AVANCE III 400 WB spectrometer (Germany) equipped with a 4 mm magic angle spinning (MAS) probe. ^29^Si MAS NMR spectra were acquired using a single pulse sequence with a 90° pulse width of 4 μs and a recycle delay of 60 s. Samples were loaded into 4 mm outer diameter ZrO_2_ rotors and spun at 12 kHz.

## 3. Results

### 3.1. Mineralogical Analysis of Sieved Excavation Soils

The potential of excavation soils as SCMs was evaluated through XRD, TGA, and XRF technologies. XRD analysis (Figure 3a) was first conducted using a zincite internal standard. The analysis identified the primary crystalline phases as quartz and kaolinite, which are indicators of potential pozzolanic reactivity with calcination. The variance in quartz and kaolinite intensity in the soils was mainly attributed to the differences in geological histories and collection depth influencing the mineral composition [43,44]. The phases for minor crystalline peaks in the XRD analysis might be related to other elements, such as silicon (Si), iron (Fe), or potassium (K), detected in the XRF analysis (Table 2). However, these phases were challenging to identify due to the low density or the absence of part of the three strongest peaks. However, their influence on the performance of the cementitious materials should be minor due to the low content and lack of pozzolanic reactivity [46,47]. Subsequent quantitative XRD revealed kaolinite mass fractions of 63.5%, 21.5%, 38.5%, and 69.9% in Soils 1 through 4, respectively.

TGA/DTG analysis in the range of 400–750 °C indicated distinct weight losses attributable to kaolinite dehydroxylation during the calcination process, as evidenced in Figure 3b,c [48]. The literature suggests that pure kaolinite undergoes a weight loss of approximately 13.76% due to dihydroxylation [49,50,51]. Corroborating XRD findings, quantitative TGA (Table 3) calculated kaolinite contents of 67.68%, 23.75%, 43.83%, and 71.41% for Soils 1–4, highlighting Soils 1 and 4 as having the highest kaolinite contents, while Soil 2 presented with the least.

XRF analysis (Table 3) exposed significant variations in the elemental compositions of the excavation soils sampled from different urban locations. Soil 2 was characterized by the highest levels of SiO_2_ and the lowest Al_2_O_3_, whereas Soil 3 contained an elevated amount of Fe_2_O_3_. In comparison, CMK showed the highest Al_2_O_3_, with minimal other elements except for SiO_2_. It was noted that the elemental compositions of Soils 1 and 4 were closely aligned with that of CMK. Given that MK (Al_2_O_3_·2SiO_2_) acts as a primary pozzolanic component in calcined soils, the significant levels of Al_2_O_3_ in Soils 1 and 4 suggested that they possessed a higher pozzolanic capacity following calcination. Regarding the other elements, such as Fe_2_O_3_ and K_2_O, their impact on the pozzolanic activity was likely to be minor, as mentioned above, and thus unlikely to drastically alter the overall performance of cementitious materials [46,47].

In summary, the quantitative XRD, TGA, and XRF analyses revealed variations in the chemical compositions of soils from excavation sites, especially concerning the content of kaolinite and quartz.

### 3.2. Mechanical Performance of Cement Mortars

Subsequent to quantifying the chemical compositions of soils from various excavation sites, the potential of employing calcined soils as SCMs was investigated by assessing the mechanical properties of mortars with partial cement replacement by calcined soils.

Figure 4a illustrates the compressive strengths of modified mortars at different curing periods. It is important to note that mortars with CMK exhibited enhanced 7-day and 28-day strengths compared to the control, despite a marginal reduction in 3-day strength. These findings demonstrated the potential of MK-enriched calcined soils as SCMs. The mechanical performance of mortars modified with calcined soils varied based on source and curing duration, with M4-modified mortars outperforming 28-day control ones and M1 and M3 achieving similar performance to the control. In contrast, M2 consistently showed the lowest strengths across all curing times compared to the control. The observed lower strength of modified mortar at an early curing age (3 days for MC and up to 7 days for others) than control mortar can be attributed to the requisite curing duration essential for pozzolanic reactions. As curing progressed, the accumulated pozzolanic reactions were anticipated to enhance mechanical performance, ultimately exceeding the controls after 28 days of curing.

Figure 4b shows the correlation between 28-day compressive strength and the primary elemental compositions of the XRF-determined excavation soils. It is important to note that a significantly positive correlation between the compressive strength of the mortar and the aluminum (Al) content of excavation soil was observed, with an increase in strength corresponding to higher Al content. Conversely, no discernible correlations were observed between the strength and the content of other elements (silicon (Si), iron (Fe), or potassium (K)) in the soils. Combining XRD and TGA results verified that the primary source of Al was kaolinite in the soils.

Figure 4c illustrates the correlation between the kaolinite content in excavation soil and the compressive strength of mortar. TGA analysis results were used here, attributable to its insensitivity to crystallinity. A positive linear relationship was revealed: as kaolinite content increased, so did the compressive strength. Similar correlations were also found in the literature [52,53]. The reason for this is that the higher the kaolinite content in the excavation soil, the more metakaolin can participate in pozzolanic reactions, leading to higher compressive strength. From the point of intersection (53.39, 36.2) of the dashed line and the fitted solid line in Figure 4c, we can also see that if the kaolinite content of the sieved excavation soil reached 53.39%, mortars with 15% substitution by the calcined soil achieved the same compressive strength (36.2 MPa) as the control mortar M0 (i.e., without calcined soil replacement). It suggested that a minimum kaolinite content of 53.39% is necessary for the performance of calcined excavation soil admixed mortar to be comparable to that of the control specimen. Overall, the potential of calcined soils as SCMs is underscored by the enhanced performance associated with their high MK content.

These results emphasize the importance of composition assessment in developing optimized cement substitutes. The established quantitative relationship between kaolinite content and mechanical strength provides a framework for selecting kaolinite-rich soils from excavation sites to achieve desired mechanical properties.

### 3.3. Microstructural Analysis of Cement Mortars

#### 3.3.1. MIP Analysis

Porosity significantly impacts the mechanical properties of cement mortar, contributing to strength and durability. Decreased porosity is often associated with a more compact internal structure, which correlates with enhanced mechanical properties [54].

MIP results (Figure 5) demonstrated that the MC sample showed optimal pore structures and had the lowest total porosity, suggesting reduced permeability and enhanced durability potential. Although Soils 1 and 4 had decent total porosities, M4 showed a notable shift towards finer pores, likely due to its higher kaolinite content enabling more extensive hydration. Interestingly, despite higher total porosity than the control, M3 showed favorable pore structure for high strength characteristics, which may explain its comparable 28-day mechanical strength to M1 (Figure 4a) despite its significantly lower kaolinite content (Table 3). These observations suggest that other soil characteristics might impact modified mortar’s mechanical performance. M2, on the other hand, exhibited the highest total porosity and a significant volume of large pores, which was consistent with its poor strength characteristics (Figure 4a).

Results showed that excavation soil kaolinite content significantly influenced mortar pore structure refinement. Higher kaolinite soils produced better structures with lower overall porosity and fewer large pores. Conversely, lower kaolinite soils like M2 generated less favorable structures with higher porosity and larger pores. Denser structures with fewer large pores potentially offer enhanced resistance to water penetration, chloride ingress, freeze-thaw damage, etc., all of which indicate the delayed degradation of mortars over time [55]. This suggests that mortars formulated with kaolinite-rich excavation soils may exhibit superior durability and long-term performance.

Therefore, the kaolinite proportion in excavation soil is critical for optimal cementitious pore structures. Enhanced structures with fewer large pores can improve durability and performance. Therefore, targeting excavation soils with adequate kaolinite contents is an effective strategy to promote their application as SCMs.

#### 3.3.2. BSE Analysis

BSE images (Figure 6a) revealed significant microstructural changes associated with the partial replacement of cement with calcined excavation soils. The introduction of MK improved pore structure densification due to enhanced particle packing and pore filling capacity. Distinct morphologies and grayscale variations were observed, with darker grey indicating the presence of unreacted MK and quartz. Mixtures with a higher MK content, such as specimens M4 and MC, exhibited a prevalence of lighter grey phases, potentially indicating the formation of aluminum-rich hydrate particles like calcium aluminosilicate hydrates (C-A-S-H) or geopolymer gels [56]. Such phases could enhance strength development and durability, with C-A-S-H potentially increasing early-age strength and geopolymer gels improving long-term strength [57]. Concentrations of inner hydration product phases were noted along the boundaries of anhydrous cement grains.

To further examine the influence of MK on hydration products, EDS analyses were performed on samples from the control (no MK), M2 (lower MK content), and M4 (higher MK content), as shown in Figure 6bThe hydration of the control samples primarily produced calcium silicate hydrate (C-S-H) and portlandite, with the integration of minor Al into the C-S-H structure. The incorporation of MK resulted in the formation of substantial aluminum-rich phases such as C_2_A_x_H_y_, including C_2_AH_8_ and, potentially, C_2_ASH_8_ (stratlingite) [58], which presented as dark grey particles in BSE images, suggesting a significant role in microstructural development. A positive correlation between these Al phases and the MK content was observed, with a higher abundance in M4 than M2. This observation underscores the kaolinite content in the excavation soil as a critical factor for forming aluminum-rich phases.

Additionally, a larger proportion of unreacted MK was identified in M4 relative to M2, indicating variable reaction extents in different soils. The consumption of portlandite by MK likely occurs through a pozzolanic reaction, where MK reacts with portlandite to form additional C-S-H and C-A-S-H phases [59]. This reaction contributes to strength development and refines the pore structure by consuming the larger portlandite crystals and producing finer hydration products [60]. The resulting denser microstructure can enhance the durability of the mortars by reducing permeability [61].

#### 3.3.3. Hydration Products in Cement Paste

To investigate the factors contributing to the mechanical performance and microstructure of calcined soils as SCMs collected from different excavation sites, XRD, FTIR, SEM, and NMR were employed to detect differences in the chemical composition and microstructural properties of the resultant hydration products.

#### 3.3.4. X-ray Diffraction

XRD analysis of cement pastes subjected to curing periods of 3, 7, and 28 days showed prominent peaks of major crystalline phases, specifically alite, portlandite, and quartz, irrespective of curing duration (see Figure 7a–c). During cement hydration, alite dissolution upon exposure to water forms C-S-H gel and portlandite as hydration products. The decreased alite intensities during curing indicated its predominant role in the early stage of cement hydration, while the increase in characteristic peaks of portlandite at 18.07°, 34.13°, and 47.12° 2θ reflected the progression of hydration. Intense invariant quartz peaks from the substantial fine sand fractions signify an inactive component unable to participate in strength development. This limitation, along with the lower measured reactivity of P2, likely contributes to its poorer mechanical performance through an inability to generate cementitious gels. XRD analysis revealed that curing impacted crystalline hydrate development, while highlighting the implications of variable soil sand contents.

At 3 days (Figure 7a), the portlandite peak intensities for the P4 and PC samples closely matched the control, suggesting comparable portlandite levels between the specimens, aside from the highest peaks in the pure control. This supports cement-dominated early hydration, with minimal pozzolanic reactivity within the first 3 days. After 7 days (Figure 7b), closer inspection of the 18.07° 2θ portlandite peak showed intensity variations indicating portlandite content differences. The control displayed the highest peaks, followed by P2, which had lower MK and, thus, more unreacted cement. In contrast, the higher metakaolin PC blend exhibited lower portlandite intensities, supporting MK consumption of portlandite via ongoing pozzolanic reactions after 7 days, which could explain PC’s superior strength. Among 28-day cured specimens, while the control still showed relatively higher portlandite peaks, P4 and PC consistently displayed lower intensities than other groups. Efficient portlandite depletion due to heightened MK pozzolanic reactivity likely clarified the excellent strength of P4 and PC. Forming late-stage hydrates instead of portlandite is known to densify cement paste, enhancing strength [62]. The variations in portlandite peak intensities observed in the XRD analysis suggest differences in the rate and level of pozzolanic reactions occurring in the mixes [63]. The lower intensities in specimens with higher MK content, especially at later curing ages, indicate a more efficient consumption of portlandite due to the increased availability of MK [59]. This accelerated pozzolanic reaction contributes to the formation of additional strength-enhancing phases and refines the pore structure, leading to improved mechanical performance, as evidenced by the higher strength of P4 and PC.

Consistent quartz peaks across specimens and curing ages underscore the stability of the substantial inert fine sand fractions. However, this sand limits strength development through the inability to form cementitious gels, explaining the lower P2 mortar properties with less favorable sand. Lower portlandite peaks in other groups, especially PC and P4, indicated SCM consumption of portlandite via ongoing pozzolanic reactions. The conversion of portlandite into additional hydrates can explain the excellent 7-day strength of PC and P4.

#### 3.3.5. Fourier Transform Infrared Spectroscopy

FTIR was employed to examine the characteristic functional groups present in the cement pastes cured for 28 days and modified with different soil samples, as shown in Figure 8. The absorption peaks observed at 3645 cm^−1^, along with a broad band ranging from 3000 to 3500 cm^−1^, attributed to O-H stretching vibrations within hydrogen-bonded water molecules, with decreasing intensities at later hydration stages [64]. Peaks corresponding to the bending vibrations of H-O-H inbound water appeared at 1637 cm^−1^ [65]. Stretching vibrations within O-C-O, attributable to atmospheric carbonation, were present at 1489 cm^−1^ on the surfaces of the paste samples [66]. The CO_3_^2−^ groups, indicative of C-O vibrations in carbonates, were responsible for the production of absorption bands at 1420 cm^−1^ [67]. Further, the presence of Si-O stretching vibrations outside the tetrahedral silicate planes relating to SO_4_^2-^ in sulfates was evidenced at 1116 cm^−1^ [68]. Intense absorption at 970 cm^−1^, assigned to asymmetric Si-O stretching within C-S-H gels, exhibited systematic shifts correlating with the variations in Ca/Si ratio [69,70,71].

Minimal variations were observed in peak positions and types between soil-modified and control pastes, implying that the incorporation of calcined soils had minimal influence on the vibrational modes of chemical groups within the cement hydrates. An exception to this trend was a reduction in O-H stretching intensity at 3645 cm^−1^ in the soil-modified samples, coupled with a new peak at ~3425 cm^−1^. These changes correlated with an increased formation of hydrates and pozzolanic reactivity, as corroborated by XRD patterns. These findings suggest favorable compatibility and enhanced reactivity from a molecular chemistry perspective. Soils contribute to supplementary hydration reactions through the consumption of portlandite, promoting densification. Meanwhile, the additional shoulder peak may indicate the changes in C-S-H hydrogen bonding consequent to Al substitution from the soil.

Moreover, no new peaks were observed in the FTIR spectra of soil-modified samples compared with the control one. Calcined excavation soils could potentially introduce impurities that negatively influence the cement products’ hydration process and performance. For example, unexpected salt may affect the cement paste’s setting time and long-term durability [72]. The absence of unexpected peaks in the FTIR spectra of soil-modified samples compared with the control indicates that no significant foreign materials were introduced by the calcined excavation soils, ensuring the safety and compatibility of the modified cement paste.

Generally, the coupled XRD and FTIR analysis provides critical information at the molecular and phase levels, thereby enabling a thorough evaluation of performance in calcined soil-modified cement pastes.

#### 3.3.6. Scanning Electron Microscope

SEM reveals the microstructural and morphological differences related to adding calcined soil to paste samples cured for 28 days (Figure 9. The control sample was characterized by the presence of smooth, hexagonal faceted portlandite crystals with well-defined edges, indicative of cement hydration and C-S-H formation. In contrast, the PC sample demonstrated a denser microstructure with reduced porosity, indicative of enhanced particle packing resulting from pozzolanic reactions between the highly reactive CMK and existing hydrates. The characteristic portlandite morphologies were absent in the PC blend, suggesting a significant reduction likely caused by the extensive involvement of CMK in secondary pozzolanic reactions.

Moreover, globular nanostructured C-A-S-H compounds were evident along phase boundaries in the soil-modified paste [73]. C-A-S-H gels generally have a more refined pore structure, and contribute to a denser microstructure within the cement matrix [74]. This densification and the formation of additional strength-enhancing phases due to pozzolanic reactions lead to improved mechanical properties such as higher compressive strength and enhanced durability in mortars containing C-A-S-H gels [75]. The observed portlandite exhibited rounded edges, reduced volumes, and pitted surfaces, indicative of a microstructure with superior mechanical performance. Since the dissolution of portlandite within samples containing calcined soil or CMK correlated directly with MK pozzolanic reactivity, such morphologies provide visual confirmation of pozzolanic reactions. Compared to the defined facets delineating portlandite in the control sample, the changes confirmed the influence of MK content of calcined soil on the microstructure-modified mortar.

The P4 and P1 samples, with higher MK and lower quartz content, demonstrated remarkably similar morphologies and characteristics. The portlandite crystals exhibited reduced volumes, rounded edges, and denser C-S-H and C-A-S-H nanostructures. In comparison, P2, with its lower MK content and higher quartz proportion, presented less refined microstructures. While larger portlandite regions remained, their boundaries were less defined than those of the control region, and overall, they lacked microstructural refinement. This aligned with the MK content governing pozzolanic reactivity and subsequent calcium hydroxide dissolution. At higher proportions like P4 and PC, increased consumption of portlandite led to both depletion and phase substitutions by additional strength-enhancing aluminosilicate hydrates. However, there was less secondary pozzolanic activity at lower MK levels, such as in P2, yielding inferior microstructural development compared to P4 or the MK samples. Thus, the proportion of reactive MK components directly influences the paste morphology, density, and mechanical properties by controlling pozzolanic reactions.

#### 3.3.7. Solid-State Nuclear Magnetic Resonance

The hydration of cement primarily yields amorphous C-S-H and C-A-S-H gels, and diffraction methods cannot easily determine their structures. ssNMR spectroscopy, however, offers crucial information on the local structural environment around isotope nuclei irrespective of crystalline periodicity. Magic angle spinning ^29^Si NMR is an essential method to probe the silicate and aluminate sites, providing insights into the degree of silicate chain connectivity (Q^i^ tetrahedra notation) along with substitution mechanisms. The resonances at approximately −71.2 ppm (Q^0^) represented an island-like silicon–oxygen tetrahedron, which means that this silicon–oxygen tetrahedron was not connected to other tetrahedra. During the hydration process of cement, this state of silicon–oxygen tetrahedra may indicate a partially reacted or partially hydrolyzed silicate [76]. The approximately −79.5 ppm (Q^1^) and −84 ppm (Q^2^) resonances represented silicon sites with one and two bridging oxygens, respectively. Their intensity variations indicated hydration progression: a decrease in Q^0^ demonstrated the hydration of cement, whereas increases in Q^1^ and Q^2^ reflected silicate chain lengthening and cross-linking due to ongoing alite reaction and C-S-H gel formation [76].

As shown in Figure 10, these resonances were predominant throughout all of the hydration ages assessed. When calcined soil or CMK was applied, a shoulder resonance between Q^1^ and Q^2^, assigned as Q^2^(1Al) and identified by a chemical shift at approximately −82 ppm, provided the first spectroscopic evidence for directly detecting Al substitution within the silicate chains. Given that Al prefers a penta-coordinated environment, in contrast to the tetrahedral coordination of silicon, its incorporation is implicated in inducing strain and chain fragmentation. An analysis of the ^29^Si NMR spectra from the control sample identified the temporal growing intensity of Q^2^(1Al), suggesting enhanced aluminate uptake over time, corroborating the transition from C-S-H to C-A-S-H gels through a substitution mechanism. This was further confirmed by the analysis of cement samples mixed with SCM: the P2, P4, and PC samples showed a slight decrease in Q^1^ resonance intensity, but a significant enhancement in the Q^2^(1Al) peak. The ssNMR, therefore, delivered vital atomic-scale clarification of the substitution mechanisms underlying the spectroscopic observations.

Deconvolution of the NMR spectra quantified the proportions of Q^0^, Q^1^, Q^2^, and Q^2^(1Al), as detailed in Table 4. Table 4 also presents the analytical results of ^29^Si NMR, highlighting the significance of the variables Ψ (mean chain length of the C-A-S-H gel, Formula (1)), Al/Si (aluminum-to-silicon ratio in the C-A-S-H gel where Al^3+^ replaces Si^4+^, Formula (2)), and α (extent of reaction of C_3_S and C_2_S in the cement paste, Formula (3)) in characterizing the hydration process [77]. After 3-day hydration, the concentration of Q^0^ species in the control cement paste remained high, comprising 47.85% of the composition. As the hydration progressed, there was a decline in the concentration of Q^0^; after 7 days, the reduction rate decreased, indicating a deceleration in the hydration reaction. Simultaneously, the Ψ, Al/Si, and α values consistently increased, implying the C-S-H gels’ chain length elongation and the increased formation of Q^2^(1Al) species, evidenced by the increased Ψ and Al/Si values in Table 4. The data align with NMR principles, where specific [SiO_4_]^4−^ configurations correlate with the resonances observed in the ^29^Si NMR spectrum. The presence of Al-substituted C-A-S-H gels in P4 and PC correlated with increased gel formation and longer chain lengths, creating a denser and more interconnected gel network to potentially improve the strength and stiffness of the cementitious materials [78]. Additionally, the incorporation of Al and Ca in calcined excavation soil enhanced chemical bonding, resulting in stronger ionic and electrostatic interactions that contributed to a more cohesive material and refined pore structure, as proved in Section 3.3.1, which could facilitate better hydration and curing [79].

Additionally, a positive correlation between the properties of 28-day hydrated cement paste—Ψ, Al/Si, and α—and kaolinite content in excavation soil was observed (Figure 11). This outcome aligned with previous assessments of mechanical performance, porosity, and morphological studies. The reason likely arises from the additional Al source provided by calcined excavation soil during hydration. This led to increased Al substitution within the C-S-H structures of P4 and PC, promoting the formation of longer and more abundant C-A-S-H chains [80]. As a result, the microstructure became denser with a refined pore size distribution, contributing to improved mechanical properties, such as strength.

In summary, MK from calcined soil contributes to hydration and influences the structural characteristics of the hydration products. The inclusion of calcined excavation soil within the cement pastes enhances the hydration process, leading to an increased quantity and average chain length of the C-A-S-H gels, potentially enhancing the cementitious materials’ mechanical strength and pore structure.
(1)$$\Psi =\frac{I(Q^{1})+I(Q^{2})+\frac{3}{2}I[Q^{2}\left(1Al\right)]}{\frac{1}{2}I(Q^{1})}$$
(2)$$Al∕Si=\frac{0.5\times I[Q^{2}\left(1Al\right)]}{I(Q^{1})+I(Q^{2})+I[Q^{2}\left(1Al\right)]}$$
(3)$$\alpha =\left[100-I\left(Q^{0}\right)\right]\times 100\%$$

## 4. Discussion

This study evaluated the suitability of recycling excavation soils from different resources as SCMs by conducting a comprehensive assessment, including quantitative mineralogical analysis of sieved soil samples, performing analyses to determine mechanical, pore structure, and morphological properties, and microscopic testing of hydration products. A higher kaolinite content was found to be the most critical criterion for assessing the potential of excavation soil as a SCM. Considering the close relationship between kaolinite content and Al element, the content of Al element (or aluminum oxide) obtained from XRF can be considered an essential indicator for briefly estimating the potential pozzolanic activity of soils.

Various mineralogical analyses were conducted to characterize four sieved soil samples sourced from four cities. XRD patterns indicated that the crystalline phases of all of the sieved excavation soil samples were dominated by quartz and kaolinite, with the latter being the requisite precursor for MK. Despite their visual similarities, the Soil 2 sample contained considerably higher quartz content than other soils, while Soil 4 presented the most intense kaolinite diffraction peaks. The kaolinite content in these soils (as quantified with TGA analysis) varied significantly from 23.75% to 71.41%, depending on the specific geographical source. Post-calcination yielded calcined soils with MK content ranging from 20.41% to 61.36%. XRF analysis revealed the difference in the elemental composition of the soils: Soil 2 contained the highest levels of SiO_2_, yet the lowest Al_2_O_3_. By contrast, Soil 4 had the highest Al_2_O_3_ content, which was essential for MK formation upon dehydroxylation. This compositional analysis identified soil samples with high kaolinite content, potentiating applications as SCMs. The significant variations in the mineralogical composition of excavation soils directly impacted the MK content of the calcined product, consequently influencing its performance in cement and concrete applications [81]. Therefore, a comprehensive mineralogical analysis of excavation soil is crucial to ensure quality and suitability before its utilization as an SCM. Additionally, research into optimizing calcination process parameters can further maximize MK yield from the calcined excavation soil.

The mechanical strength of mortars modified by calcined excavation soil showed an excellent linear correlation with kaolinite content and a slightly better one with Al_2_O_3_ content over various curing durations. A higher kaolinite content, or more specifically, a higher Al element content, is the most critical criterion for assessing the potential of calcined excavation soil as an SCM. Soil 4, with the highest kaolinite content (and Al_2_O_3_ content), exhibited superior compressive strength compared to Soil 2. The established quantitative relationship between kaolinite content (or Al_2_O_3_ content) and mechanical strength provides a framework for selecting kaolinite-rich soils from excavation sites to achieve desired mechanical properties. It can be derived from the linear regression curve that 15% of cement can be replaced with calcined excavation soil (containing 53.39% kaolinite) without sacrificing mechanical performance. These results emphasize the importance of composition assessment, particularly the kaolinite proportion (and Al_2_O_3_ content), for developing optimal cement substitutes.

An increase in kaolinite content in excavation soil enhanced not just compressive strength but also the pore structure of the mortar. MIP analysis of mortar samples revealed that using soils with higher kaolinite content resulted in lower overall porosity and densified morphology with fewer large pores than soils with less kaolinite. This was further verified using BSE imaging, which showed enhanced densification and pore-filling capabilities, verifying the role of MK in calcined excavation soil in enhancing particle packing over plain cement. The kaolinite proportion in excavation soil is critical for optimal cementitious pore structures. Targeting excavation soils with adequate kaolinite contents is an effective strategy to promote their application as SCMs.

Through the analysis of hydration products using XRD, FTIR, SEM, EDS, and solid-state NMR techniques, microstructural changes in cement hydration products were observed when calcined excavation soil was incorporated as an SCM. This can be attributed to the pozzolanic reactivity of the calcined excavation soil, which primarily stems from the presence of metakaolin. OPC undergoes hydration to produce C-S-H and calcium hydroxide, which, in turn, reacts with the metakaolin in the calcined soil through a secondary hydration process known as the pozzolanic reaction [36]. This reaction leads to the formation of additional C-S-H and C-A-S-H, contributing to the observed microstructural changes and influencing the overall performance of the cement matrix [73,82]. Al substitution occurred within the C-S-H gel, forming C-A-S-H gels with longer chain lengths. This resulted in a denser and more interconnected gel network, enhancing the strength and stiffness of the cement materials [78].

## 5. Conclusions

This study examines the key factors affecting the recycling of excavation soils as SCMs from various resources. The primary conclusions drawn are as follows:(1)The kaolinite content in excavation soils varies significantly depending on their source, which has crucial implications for their practical use as SCMs after calcination.(2)A linear correlation was found between kaolinite content in excavation soils and the compressive strength of cement mortars. It is estimated that mortars prepared with calcined excavation soils containing a kaolinite content exceeding a specific value (herein 53.39%) can achieve 28-day compressive strength equivalent to or surpassing that of plain cement mortar. However, the actual value might vary when more soil samples are involved since this might influence the fitting.(3)Higher kaolinite content in excavation soil correlates with a more refined pore structure, characterized by reduced total porosity and the enhanced distribution of tiny pores, which is crucial for improving cementitious materials’ durability and mechanical strength.(4)The inclusion of calcined excavation soil within cement paste also promotes the pozzolanic reaction, leading to an increased aluminum-to-silicon ratio in the C-A-S-H gel, an extended average chain length of the C-A-S-H gels, and the accelerated hydration of OPC within the cementitious materials.(5)Kaolinite content is the most critical criterion for assessing the potential of excavation soils as SCMs. XRF, a frequently used method to characterize the composition of soils, provides the content of Al element (or aluminum oxide), which is considered an important indicator of soil potential pozzolanic activity.

## Figures and Tables

**Figure 1 materials-17-02289-f001:**
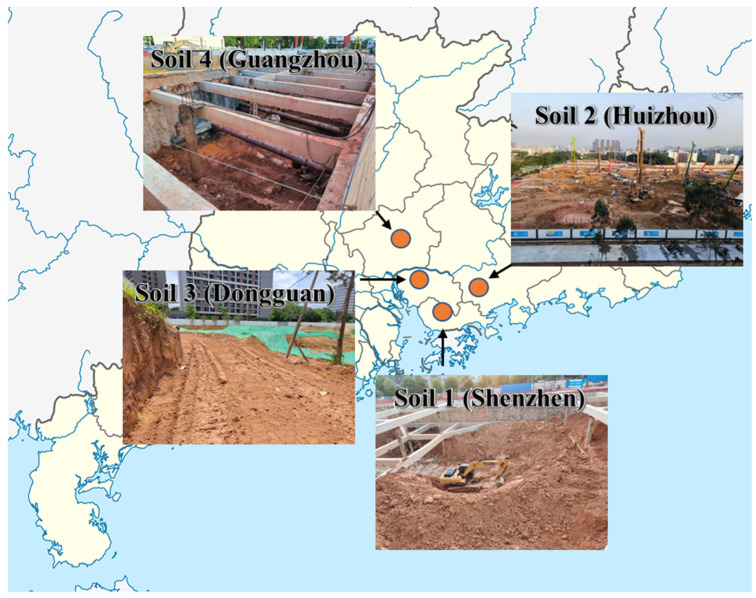
Collection locations and sites of excavation soils.

**Figure 2 materials-17-02289-f002:**
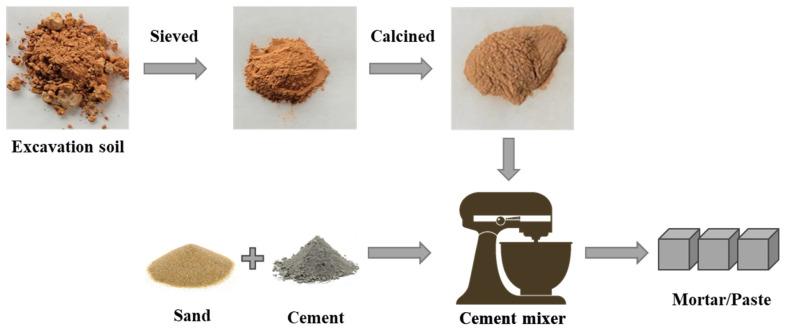
Schematic diagram of excavation soil treatment and application as cement substitutes.

**Figure 3 materials-17-02289-f003:**
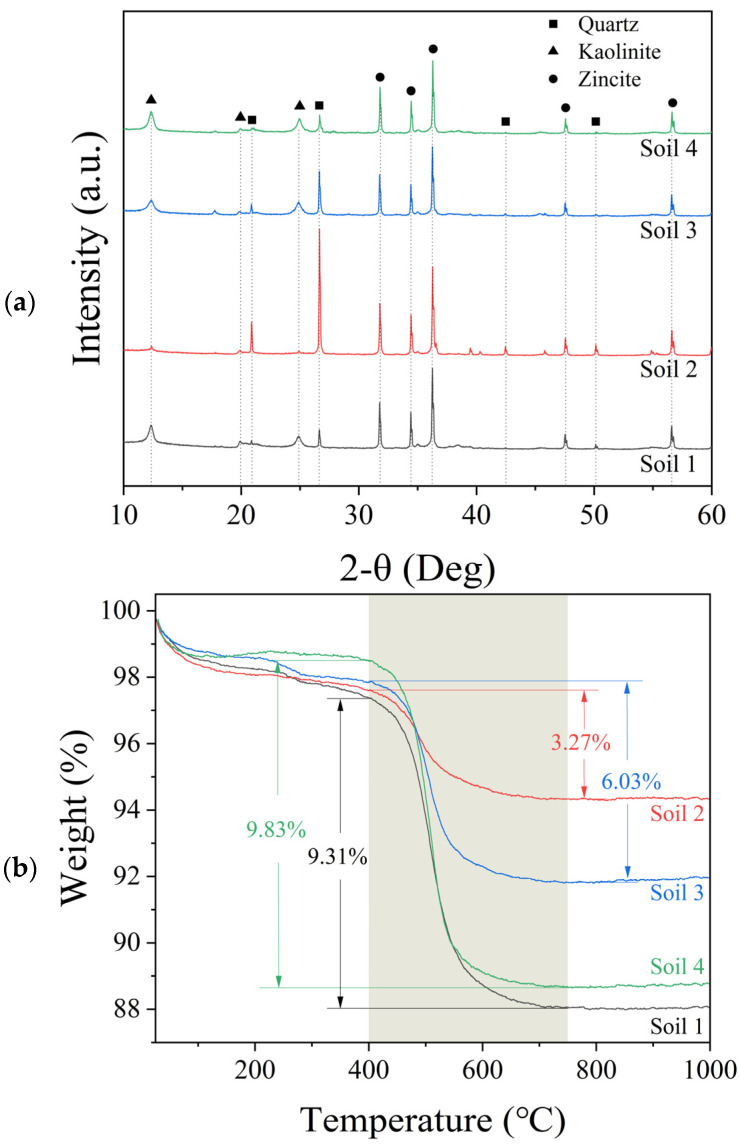
Spectral analysis of sieved excavation soils: (**a**) XRD, (**b**) TGA, and (**c**) DTG.

**Figure 4 materials-17-02289-f004:**
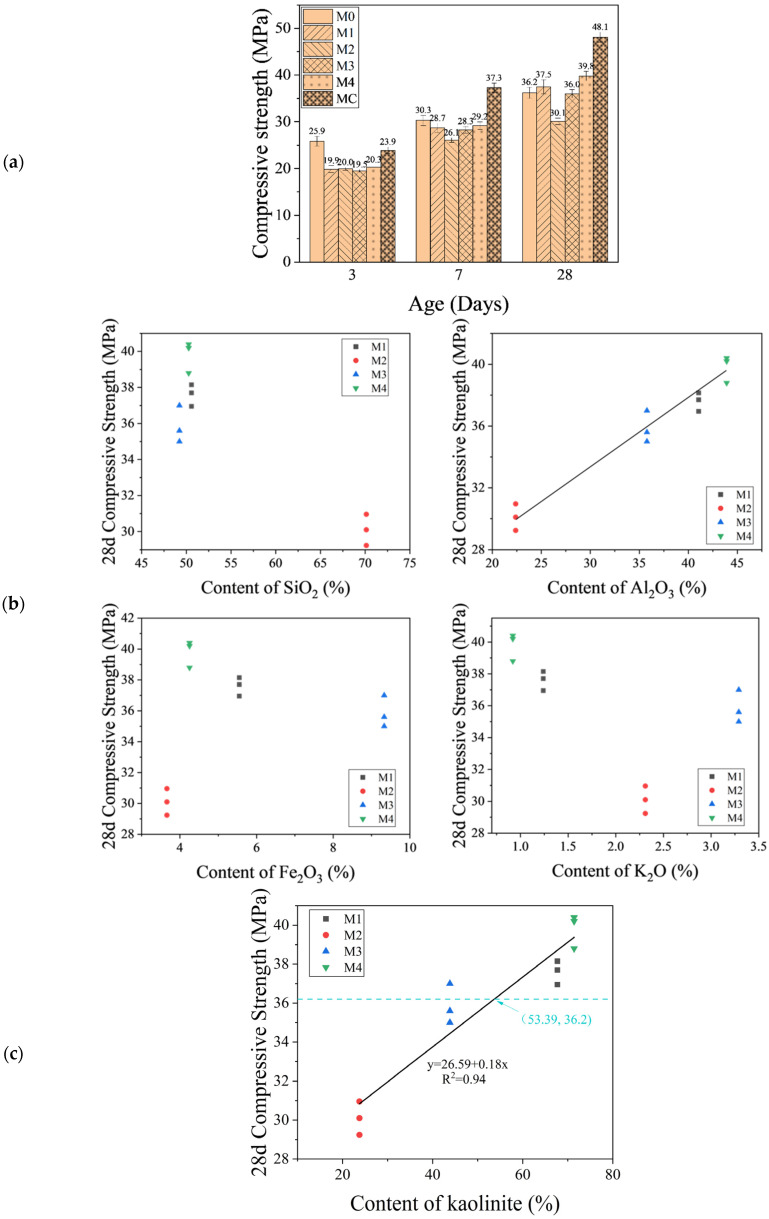
Compressive strength of mortars. (**a**) Illustration of mortars’ compressive strength after 3, 7, and 28 days of curing. (**b**) Relationship between the 28-day compressive strength and the elemental compositions. (**c**) Correlation between the 28-day compressive strength and kaolinite content in excavation soils. The blue dashed line indicates the 28-day compressive strength of specimen M0 (no calcined excavation soil involved). The coordinates (53.39, 36.2) represent the point of intersection of this dashed line and the fitted solid line.

**Figure 5 materials-17-02289-f005:**
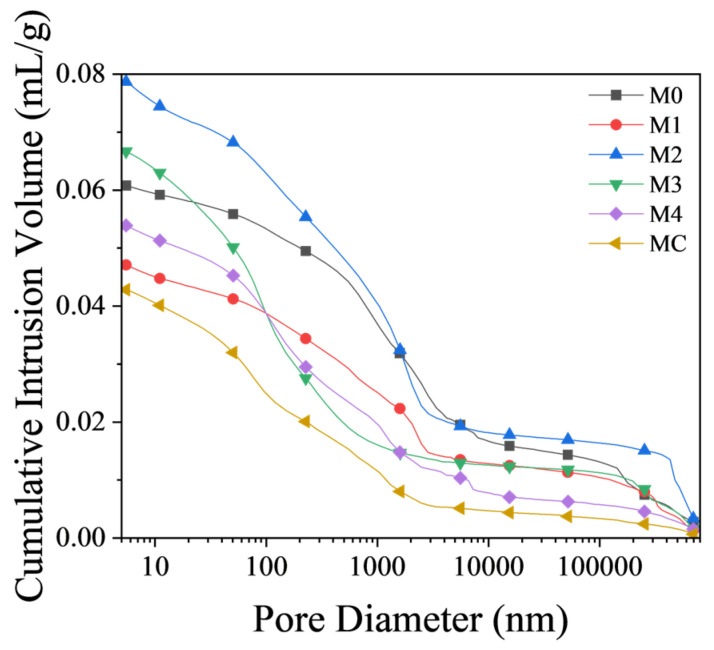
Pore size distribution in cement mortars.

**Figure 6 materials-17-02289-f006:**
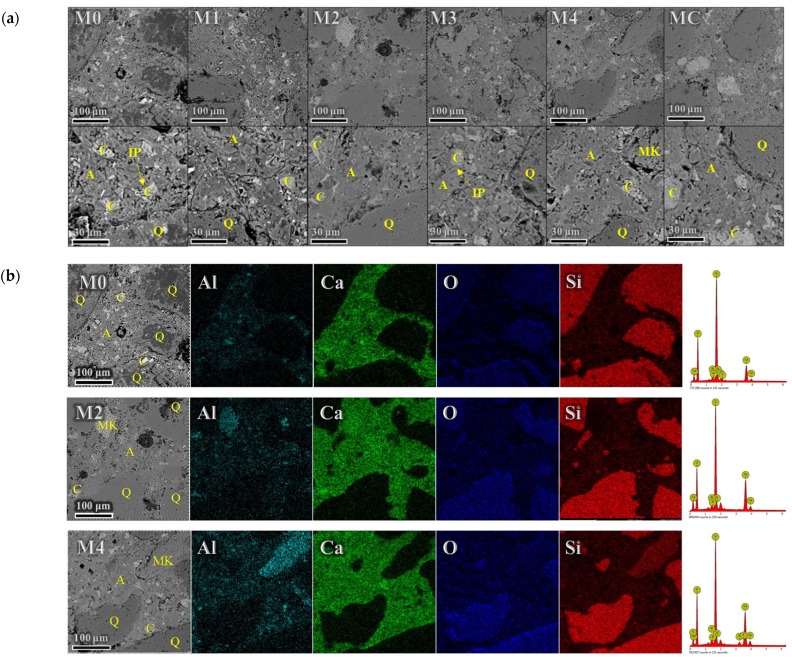
BSE (**a**) and EDS (**b**) micrographs of mortars cured for 28 days. A: Aluminum-bearing phases, C: Cement clinkers, IP: Inner products, MK: Unreacted metakaolin, Q: Quartz.

**Figure 7 materials-17-02289-f007:**
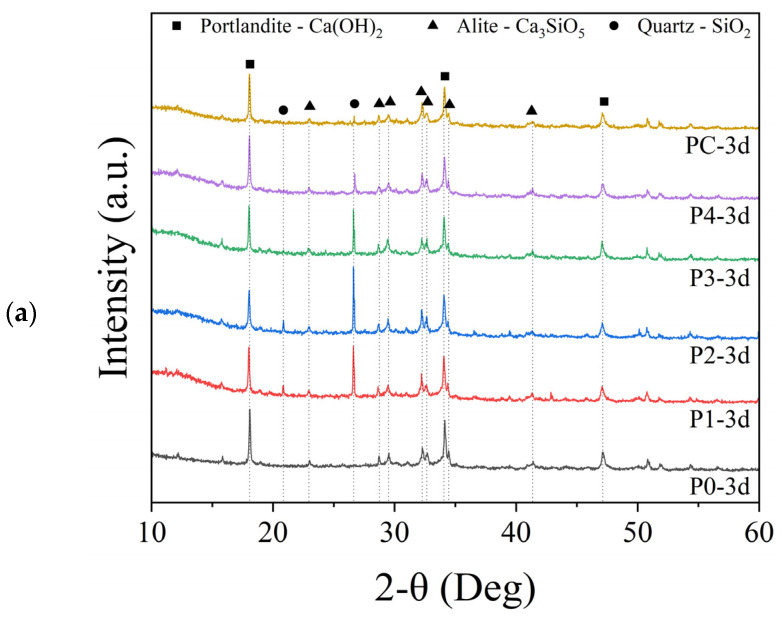
XRD spectra of cement paste samples at curing durations of (**a**) 3, (**b**) 7, and (**c**) 28 days.

**Figure 8 materials-17-02289-f008:**
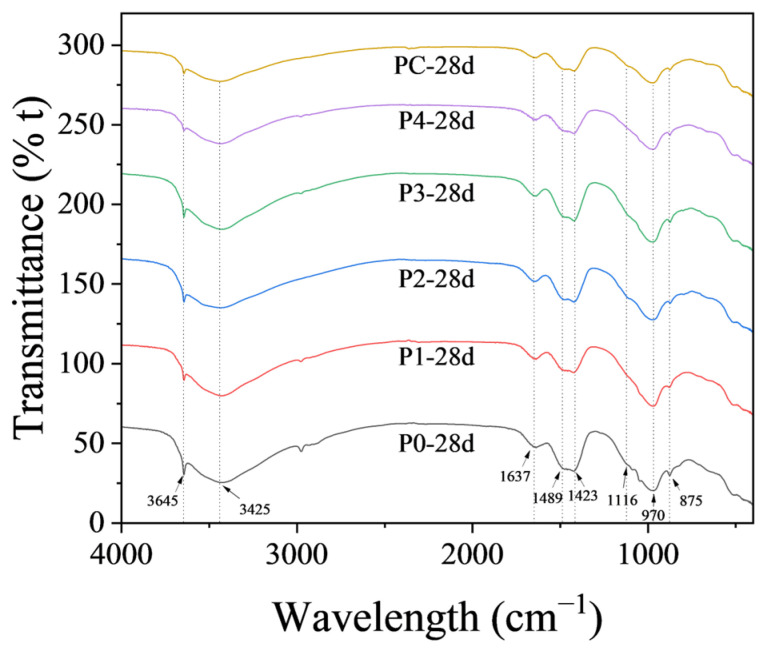
FTIR spectra of cement paste samples.

**Figure 9 materials-17-02289-f009:**
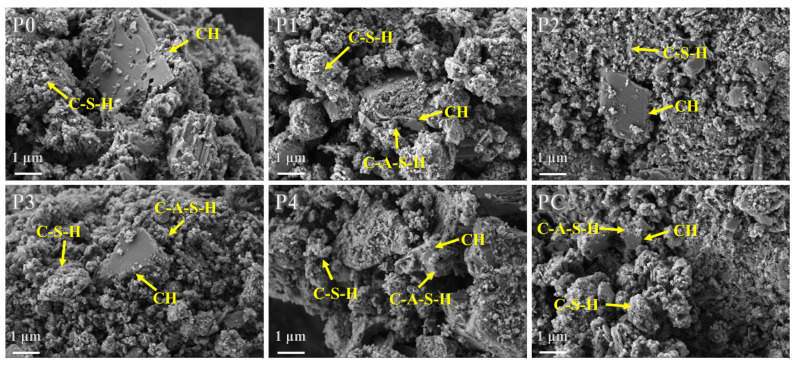
SEM images of cement paste with calcined soil and CMK as SCMs (scale bar 1 μm).

**Figure 10 materials-17-02289-f010:**
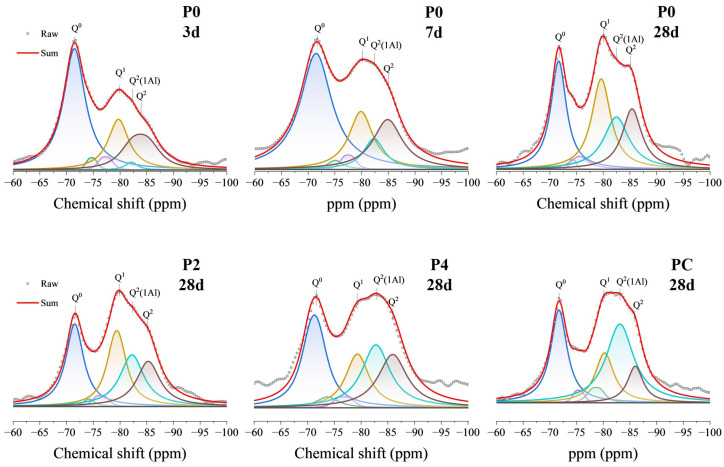
^29^Si NMR spectra with corresponding deconvolutions and fits for the cement hydrates at different curing ages. The original spectrum is shown in red. Deconvoluted peaks are highlighted in different colors for clarity: Blue represents Q^0^, Yellow denotes Q^1^, Cyan indicates Q^2^ (1Al), and Brown marks the remaining Q^2^ peaks.

**Figure 11 materials-17-02289-f011:**
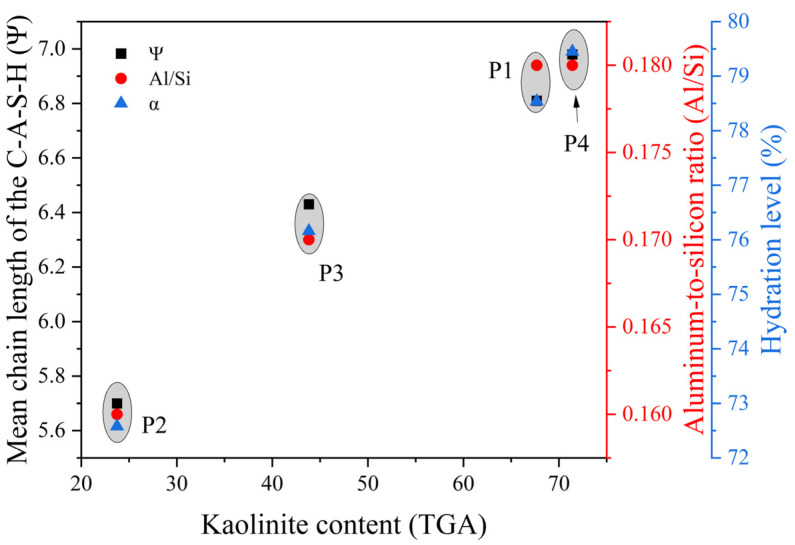
Relationship between kaolinite content in excavation soil and the parameters Ψ, Al/Si, and α in the kaolinite modified mortar.

**Table 1 materials-17-02289-t001:** Chemical composition and physical properties of standard cement.

Chemical Composition	Mass Percent (%)
CaO	64.68
SiO_2_	21.77
Al_2_O_3_	4.62
Fe_2_O_3_	3.62
MgO	2.80
SO_3_	0.46
Na_2_O equivalent	0.502
*f*-CaO	0.92
Specific gravity	3.15
Specific surface area (m^2^/kg)	343

**Table 2 materials-17-02289-t002:** Oxide-based chemical composition of sieved excavation soils and CMK, determined by XRF analysis.

Chemical Composition (wt%)	SiO_2_	Al_2_O_3_	Fe_2_O_3_	K_2_O	CaO	TiO_2_	MgO	SO_3_	Na_2_O	L.O.I
Soil 1	44.54	36.15	4.89	1.09	0.50	0.46	0.17	0.09	0.04	11.94
Soil 2	66.14	21.12	3.45	2.18	0.16	0.43	0.55	0.07	0.06	5.69
Soil 3	45.27	32.90	8.58	3.03	0.15	1.14	0.47	0.11	0.06	8.04
Soil 4	44.62	38.98	3.77	0.82	0.02	0.29	0.06	0.09	-	11.23
CMK	53.43	43.22	1.12	0.20	0.08	1.12	0.02	-	0.05	-

**Table 3 materials-17-02289-t003:** Results of quantitative TGA analysis for kaolinite content in sieved excavation soils.

Mass Ratio (wt%)	Soil 1	Soil 2	Soil 3	Soil 4
Lost between 400 °C and 750 °C	9.31	3.27	6.03	9.83
Kaolinite content	67.68	23.75	43.83	71.41
MK in calcined soils	65.56	23.00	42.45	69.17

**Table 4 materials-17-02289-t004:** Analytical results of ^29^Si NMR.

Sample	Curing Age/Days	The Cumulative Integrated Intensity/%				Ψ	Al/Si	α/%
		I (Q^0^)	I (Q^1^)	I (Q^2^)	I (Q^2^(1Al))			
Control	3	52.15	21.87	26.78	1.89	4.71	0.02	47.85
	7	49.22	23.49	20.42	8.90	4.88	0.08	50.78
	28	27.01	33.78	20.15	22.56	5.20	0.15	72.99
P1	28	21.46	26.14	22.53	26.86	6.81	0.18	78.54
P2		27.42	30.71	20.54	24.19	5.70	0.16	72.58
P3		23.84	27.59	22.31	25.87	6.43	0.17	76.16
P4		20.55	25.68	23.49	26.94	6.98	0.18	79.45
PC		18.53	22.64	10.93	41.75	8.50	0.28	81.47

## Data Availability

Data are contained within the article.

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
