# Peer review of "Recycled Excavation Soils as Sustainable Supplementary Cementitious Materials: Kaolinite Content and Performance Implications"

_materials, 2024, doi:10.3390/ma17102289_

Round 1
Reviewer 1 Report
Comments and Suggestions for Authors
The manuscript presents a comprehensive study on the potential of repurposing calcined excavation soils as sustainable supplementary cementitious materials (SCMs). The authors have conducted a thorough investigation, including mineralogical analysis, mechanical performance evaluation, microstructural characterization, and analysis of hydration products. The findings suggest that excavation soils with a kaolinite content above 53.39% can produce mortars of equal or superior quality to plain cement mixes due to refined pore structures, microstructural densification, and enhanced hydration reactions.
The study is well-structured, and the experimental approach is rigorous, involving various characterization techniques such as X-ray diffraction (XRD), thermogravimetric analysis (TGA), X-ray fluorescence (XRF), mercury intrusion porosimetry (MIP), scanning electron microscopy (SEM), and solid-state nuclear magnetic resonance (ssNMR). The results are well-presented and supported by relevant literature citations.
Overall, the manuscript provides valuable insights into the potential of excavation soils as sustainable SCMs, highlighting the importance of kaolinite content and aluminum content as critical indicators. The findings contribute to sustainable construction practices by proposing a promising avenue for recycling excavation soils, thereby reducing environmental impact and advancing sustainable construction.
Comments
- The introduction section provides a good overview of the topic and the motivations behind the study. However, it would be beneficial to include a brief discussion on the potential economic and environmental benefits of using calcined excavation soils as SCMs compared to traditional SCMs.
- In the Materials section (Section 2.1), the authors mention sourcing excavation soils from four major cities in southern China. It would be helpful to provide more information on the specific locations and characteristics of the excavation sites, as soil properties can vary significantly even within a small geographical area.
- The authors mention using a 150-μm sieve for the sieving process (Section 2.2). Could you provide a justification for choosing this particular sieve size? Additionally, it would be interesting to explore the potential effects of different sieve sizes on the performance of calcined excavation soils as SCMs.
- The statement "Despite recognized geographical variability in kaolinite content, existing literature lacks a systematic analysis across diverse soil deposits on reactivity and mechanical properties within cementitious binders" (Page 1, Lines 21-23) raises a question: Are there any specific reasons or factors contributing to this lack of systematic analysis in the literature?
- In the XRD analysis (Section 3.1), the authors mention identifying quartz and kaolinite as the primary crystalline phases. However, it would be valuable to discuss the potential presence and implications of other minor crystalline phases, if any.
- The authors state that "Soil 2 exhibited the highest quartz intensities, followed by Soil 3" (Section 3.1). Could you provide a potential explanation or hypothesis for the observed differences in quartz intensities among the soil samples?
- The statement "Literature suggest that pure kaolinite undergoes a weight loss of approximately 13.76% due to dehydroxylation" (Section 3.1, Page 4, Line 151) requires a citation to support this claim.
- In the XRF analysis (Section 3.1), the authors mention that Soil 4 had the highest Al2O3 content, which is essential for MK formation. However, it would be beneficial to discuss the potential effects of other oxides, such as Fe2O3 and K2O, on the performance of calcined excavation soils as SCMs.
- The mechanical performance assessment (Section 3.2) shows a linear correlation between kaolinite content and compressive strength. Could you provide insights into the potential mechanisms or reactions that contribute to this observed correlation?
- The authors state that "MIP analysis of mortar samples revealed that using soils with higher kaolinite content resulted in lower overall porosity and densified morphology with fewer large pores than using soils with lower kaolinite" (Section 3.3.1). It would be interesting to explore the potential implications of this observation on the durability and long-term performance of the modified mortars.
- In the BSE analysis (Section 3.3.2), the authors mention observing dark grey particles corresponding to aluminum-rich phases. Could you provide more details on the specific aluminum-rich phases identified and their potential implications for the mechanical and durability properties of the mortars?
- The statement "EDS observations of declining calcium corresponding to MK's consumption of portlandite" (Section 3.3.2) raises a question: Could you elaborate on the potential mechanisms involved in the consumption of portlandite by MK and its impact on the microstructure and properties of the modified mortars?
- The XRD analysis of cement pastes (Section 3.4.1) revealed that the intensity of portlandite peaks varied with curing age and the presence of calcined soils or CMK. Could you provide insights into the potential reasons for these variations and their implications for the hydration process and mechanical performance?
- The authors mention the formation of C-A-S-H gels due to the incorporation of MK (Section 3.4.1). Could you discuss the potential differences in the properties and performance of C-A-S-H gels compared to the traditional C-S-H gels, and how these differences might influence the overall behavior of the modified mortars?
- In the FTIR analysis (Section 3.4.2), the authors state that "the similarities of functional group in FTIR spectra of both soil-modified and control samples indicate that calcined soils did not introduce harmful or unexpected materials" (Page 13, Lines 297-299). Could you provide more details on the potential harmful or unexpected materials that might be introduced by calcined soils, and how the FTIR analysis helps to rule out their presence?
- The SEM analysis (Section 3.4.3) revealed the presence of globular nanostructured calcium aluminosilicate hydrates (C-A-S-H) along phase boundaries in the soil-modified pastes. Could you discuss the potential implications of these nanostructured hydrates on the mechanical and durability properties of the modified mortars?
- In the ssNMR analysis (Section 3.4.4), the authors mention the detection of aluminum substitution within the silicate chains, leading to the formation of C-A-S-H gels. Could you provide more insights into the potential effects of aluminum substitution on the structure and properties of the C-A-S-H gels, and how these effects might influence the overall performance of the modified mortars?
- The authors state that "a positive correlation between 28 days hydrated cement paste properties - Ψ, Al/Si, and α – and kaolinite content in excavation soil was observed (Figure 11)" (Section 3.4.4). Could you discuss the potential reasons behind this positive correlation and its implications for the mechanical and microstructural properties of the modified mortars?
- In the Discussion section (Section 4), the authors highlight the kaolinite content and aluminum content as critical indicators of excavation soil viability for SCM application. Could you suggest potential strategies or methods for efficiently assessing the kaolinite and aluminum content of excavation soils before their use as SCMs, considering the geographical variability mentioned in the manuscript?
Reviewer 2 Report
Comments and Suggestions for Authors
The work presented by the authors is considered interesting for the readers and explores new possibilities for the improvement of common mortars and cements.
However, some deficient aspects found during the review of the manuscript should be improved.
1. Abstract: A brief mention of the name of the tests conducted in this research is recommended. The ages of the mortars investigated should also be mentioned.
2. Section 2. Change "Experimental approach" to "Materials and methods".
3. Lines 76 to 80. The text is centred, it must be justified.
4. Line 80. Delete the parenthesis at the end of Line 80.
5. Figure 1 has to be cited after Table 1.
6. Subsection 2.3. Change "Testing" to "Methods".
7. Lines 155 to 156. Put both lines at the same level.
8. Table 3. The loss on ignition (LOI) has to be provided.
9. Section 4 Discussion. The authors need to discuss the results further and compare them with the traditional results of other researchers. This section requires a large number of citations which are absent here. Please add.
Comments on the Quality of English LanguageSome small errors need to be corrected
Reviewer 3 Report
Comments and Suggestions for Authors
The work concerns research on the use of calcined soils containing kaolin as a partial replacement for cement in mortars and pastes. This topic is also consistent with sustainable development. The scope of research is sufficient. The results are presented correctly. To improve the quality of the manuscript, the following corrections should be made.
Line 57: What does LC3 mean?
Table 1: Unit missing for "specific gravity" parameter.
Line 91: Please write whether the sand:cement 3:1 proportions are by weight or volume.
Line 92: "Standard conditions" – please write approximately the temperature and RH.
Subsection 2.2 - Please write the order in which the ingredients were added to the mixer and how long mixing took.
Line 100: "The mechanical properties" – what exactly?
Subsection 2.3: Please write which tests concerned pastes and which ones concerned mortars. Only mortars are mentioned.
Fig.4a - some numbers appeared in the background on the vertical axis (probably mistake). There is an incorrect caption in this figure because the graph covers different ripening periods and not just 28d.
Fig. 4b - why is mortar with the addition of commercial metakaolin missing from the charts?
Fig. 4d - please write what the horizontal dashed blue line and the values in brackets mean.
Fig. 9 – Font colors in photos are illegible.
Section 4: Discussion. This chapter is too short and too general. In fact, it should be called "results" because it contains observations about trends in the results obtained. There is no attempt to find the causes of such trends. There are no references to literature. The use of metakaolin as a cement substitute is well known. It is therefore worth citing some examples of research results from the literature. Then it will be a discussion.
Reviewer 4 Report
Comments and Suggestions for Authors
The article is quite interesting and deals with a topic that is up-to-date. The authors followed a rather extensive programme of research. Perhaps even too extensive, as some of the studies did not add anything new to the results, and the subsequent confirmation of the same conclusions at the end got a bit tedious. The conclusions are not revealing and coincide roughly with what is already known about the use of calcined aluminas as SCMs.
There are some inconsistencies in the content of the article. The authors first mention silica as an indicator of potential pozzolanic activity, and then write that sand (because this is how they, incorrectly, refer to the quartz fraction present in the clay) is an inert material and even interferes with achieving greater strength and durability.
My doubts are raised by the fact that the authors calcined the soil samples at a high temperature (as for kaolinite) and for as long as two hours. It is possible that this irretrievably reduced the pozzolanic activity of the materials obtained, as evidenced by the significantly lower results obtained with Soil 4 despite its higher alumina content. At this point, it is worth criticising the authors' view that it is primarily the aluminium oxide content that is the determinant of the potential pozzolanic activity of soil. No, this determinant is primarily the content of specific minerals, in this case kaolinite, whose specific structure and mechanism of deformation at high temperature determines the formation of amorphous phases characterised by pozzolanic activity. The above statements should be taken into account in the article by reformulating, among other things, one of the conclusions.
I could make a few more substantive criticisms of the article, but their rank would be low and the charges themselves would be of a small calibre, so I will allow myself to omit them. I do not think anyone will take the reviewed article as a benchmark of knowledge on the study of calcine clays in a situation where there is a very rich literature available on these subjects. Below I give in points some of the more minor shortcomings that I noticed while reading the article.
1. In line 64, the word kaolinite should be written in lower case.
2. In line 276, please replace the word ‘excellent’ with another, neutral word.
3. Please provide information on the magnification at which the SEM images were taken.
4. In line 361, it should be 3 instead of 4 (there are only three formulas).
5. In conclusion 2, please use the past tense, as this is the result obtained in the studies described in the article, but the figure given does not necessarily have the value of generality.
6. In line 431, please replace the word 'hydration' with 'pozzolanic'.
7. It does not seem reasonable to me to mention belite in conclusion 5 when it is not mentioned in the text of the article.
Round 2
Reviewer 1 Report
Comments and Suggestions for Authors
The manuscript has been well-revised and is ready to move on to the next phase of publishing.
Reviewer 3 Report
Comments and Suggestions for Authors
Comments have been adequately responded to. The quality of the manuscript has improved. I have no further comments.